# Gallium(III)- and Indium(III)-Containing Ionic Liquids as Highly Active Catalysts in Organic Synthesis

**DOI:** 10.3390/molecules28041955

**Published:** 2023-02-18

**Authors:** Justyna Więcławik, Anna Chrobok

**Affiliations:** Department of Chemical Organic Technology and Petrochemistry, Faculty of Chemistry, Silesian University of Technology, Bolesława Krzywoustego 4, 44-100 Gliwice, Poland

**Keywords:** ionic liquids, organometallic catalysts, gallium-based ionic liquids, indium-based ionic liquids

## Abstract

The chemical industry still requires development of environmentally friendly processes. Acid-catalysed chemical processes may cause environmental problems. Urgent need to replace conventional acids has forced the search for sustainable alternatives. Metal-containing ionic liquids have drawn considerable attention from scientists for many years. These compounds may exhibit very high Lewis acidity, which is usually dependent on the composition of the ionic liquid with the particular content of metal salt. Therefore, metal-containing ionic liquids have found a lot of applications and are successfully employed as catalysts, co-catalysts or reaction media in various fields of chemistry, especially in organic chemistry. Gallium(III)- and indium(III)-containing ionic liquids help to transfer the remarkable activity of metal salts into even more active and easier-to-handle forms of ionic liquids. This review highlights the wide range of possible applications and the high potential of metal-containing ionic liquids with special focus on Ga(III) and In(III), which may help to outline the framework for further development of the presented research topic and synthesis of new representatives of this group of compounds.

## 1. Introduction

Ionic liquids (ILs) are organic salts that consist entirely of ions, formed by combining organic cations and inorganic or organic anions, resulting in compounds presenting melting points below 100 °C. The most significant ILs properties, such as negligible vapour pressures and very wide liquidus range, often greater than 200 °C, are of high interest to organic chemists. Furthermore, ILs have the ability to dissolve many organic and inorganic compounds and hence are frequently used as solvents [1,2,3,4,5]. The main problem of many crucial chemical processes arises from the use of hazardous acids as catalysts. Conventional approaches are associated with some issues, e.g., generation of a large amount of waste. Therefore, a growing interest in research concerning alternative catalysts, which should be active, selective and stable as well as solve problems resulting from the use of conventional acids, is of high importance [6,7,8]. On the other hand, salts of various metals, such as boron, aluminium, gallium, indium or noble metals, e.g., copper, are also successfully used in catalysis [9,10,11,12,13,14], especially in the form of halometallates, because they represent high Lewis acidity, which distinguishes them from other groups of salts. The subjects of this review are metal-based ionic liquid focusing on gallium(III) and indium(III) and their performance in organic chemistry.

Compounds containing gallium and indium, especially halides and salts, are known as substitutes for standard Lewis acids, and their use has been proven in countless chemical transformations in the field of organic chemistry. Due to remarkable catalytic properties resulting from high acidity, they can be used successfully in various chemical processes, such as Friedel–Craft reactions [15,16,17], cycloaddition reactions [18,19,20,21,22], olefin metathesis [23], various coupling reactions [24,25,26,27,28,29,30,31] and many others [12,32,33,34,35,36,37]. Although many applications of these compounds have already been explored, their usefulness extends in connection with the possibility of incorporating gallium and indium into structures of ILs with unique properties. The group 13 metals, apart from gallium and indium, also include aluminium and thallium. The existence of Tl-based ILs has not yet been reported, but aluminium-based ILs are a very frequent and widely used group of organometallic catalysts. Publications and reviews detailing their properties [38,39] as well as a possible range of applications have been already extensively studied [40,41,42,43,44,45,46]. The ILs based on Al(III) are excellent catalysts that have proven to be efficient in numerous reactions, providing high conversions, yields and selectivities. They are used in processes, such as alkylation [47,48,49], acylation [50], cycloaddition [51], esterification [52], oligomerization [53,54,55] and many other chemical transformations [56,57,58,59,60,61]. Some of them have even been implemented in the chemical industry [62,63,64]. In group 13 of the periodic table, there is also boron representing semimetals. Boron-based ionic liquids are a subgroup of ILs that have been investigated for a range of applications, including catalysis, energy storage and conversion, and separation processes [65,66,67,68,69]. Despite the potential benefits, there are also some challenges associated with the use of boron-based ionic liquids, such as their high viscosity and others; nevertheless, ongoing research efforts aim to overcome these limitations and further expand the use of boron-based ionic liquids in various fields [70].

On the other hand, ILs based on gallium and indium are of increasing interest and open a new toolbox with tunable acidic catalysts. Accurate speciation of such compounds has already been performed [71,72], while their range of applications has not yet been summarized in a review paper up to now. This work focuses on presenting their catalytic abilities in the field of organic chemistry.

The ionic liquids described in this review are generally synthesized via mixing the organic halide salt with the appropriate metal halide in a certain molar ratio [38]. Afterward, the speciation involves physicochemical characterization by a wide range of technics [38,41,71,72,73] such as multinuclear NMR spectroscopy, infrared spectroscopy or Raman spectroscopy, mass spectrometry, sometimes X-ray spectroscopy (XPS or XAS) and UV–VIS spectroscopy. To determine the thermal properties of ionic liquids, differential scanning calorimetry or thermogravimetric analysis have to be performed. Subsequently, crucial for this group of ILs is their acidity determined based on acceptor properties, as described in the next section [39,45]. The individual properties of metal-based ILs depend not only on the metal character but also on the form of cation and anion formed and hence the interionic interactions [74,75]. The molar ratio in which substrates were used also plays a significant role, especially taking into consideration the acidity of the system. The halometallate ILs comprising group 13 metals, mostly in the form of chloride, were summarized to exhibit catalytic ability resulting from the anion speciation. The catalytic properties are a direct outcome from anionic polynuclear halometallate species formation ([M_x_Cl_y_]^z−^). In chloroaluminate(III), IL anions such as Cl^−^, [AlCl_4_]^−^, [Al_2_Cl_7_]^−^, [Al_3_Cl_10_]^−^ and [Al_4_Cl_13_]^−^ exist. In the chlorogallate(III) systems, analogue anions such as Cl^−^, [GaCl_4_]^−^, [Ga_2_Cl_7_]^−^ and [Ga_3_Cl_10_]^−^ were found, while in chloroindate(III) ionic liquids only, [InCl_6_]^3−^, [InCl_5_]^2−^ and [InCl_4_]^−^ anions were reported. The occurrence of various anionic species is dependent on metal chloride molar fraction (*χ*MCl_3_) in IL composition, as described in the next section [41,71]. Regarding the physical state of halometallate ionic liquids, they can also be divided based on different metal concentrations in the ILs mixtures. Chloroaluminates(III) create homogenous ionic liquids in compositions where *χ*AlCl_3_ is less than or equal to 0.67. However, for higher values of *χ*AlCl_3_, aluminium(III) chloride was precipitated. In contrast, chlorogallate(III) Ils form homogenous liquids across the entire range studied *χ*GaCl_3_ from 0.25 to 0.75. When it comes to chloroindate(III) Ils, clear ionic liquids were only observed when *χ*InCl_3_ was less than or equal to 0.50, while beyond this composition, the mixtures were biphasic with solid particles suspended within the ionic liquid [45,71,72].

## 2. Acidity by Gutmann Acceptor Number

The Gutmann Acceptor Number (AN) is a well-known quantitative technique for measurement of acidity of compounds that are off the pH scale or inadequate for Hammett acidity function. The AN method is based on ^31^P NMR chemical shift measurements for samples containing triethylphosphine oxide (TEPO) dissolved in the tested substance. The complexation of TEPO with acid influences the chemical shift in the ^31^P NMR spectrum, based on which the acceptor number is determined. Using this method, the AN of hexane was fixed to be zero, and the acidities for other solvents and classical acids were determined from this value, as shown in Table 1 [76].

AN measurements are regularly performed for Lewis acid-based ILs to determine their acidic properties. For chlorometallate ILs, the commonly used citation is 1-octyl-3-methylimidazolium chloride ([C_8_mim]Cl) with adequate metal chlorides in various compositions *χ*MCl_3_ (*χ* = molar fraction) from 0.25 to 0.75 [29,30]. The obtained results are presented in Table 2. The ILs based on chlorogallate(III) show the highest AN among the tested ILs. ILs with *χ*GaCl_3_ = 0.75 exhibit acidity comparable to AN of trifluoroethanoic acid. On the other hand, chloroindate(III) systems were reported to be mild Lewis acids or even characterized as neutral. After all, one of the most interesting aspects of the acidity of the measured chlorometallate(III) ILs is their function as Lewis acids in neutral compositions (*χ*MeCl_3_ = 0.50), which is described below. Considering their better hydrolytic stability than aluminium-based systems, chlorogallate(III) and chloroindate(III) ILs may have a strong advantage in catalysis, where highly acidic or medium acidic conditions are required [38,39].

The ILs based on metal chlorides are a well-known group characterised with strong Lewis acidity. The strength of the Lewis acid depends on the radius of the metalloid; thus, for a series of chlorometallates (MCl_3_) containing metals from Group IIIA, their strength will increase in the following order: In < Ga < Al. The influence of chosen metal on acidity of ILs was frequently confirmed, e.g., Angueira et al. determined this dependence determining activity of ILs [C_6_mim][Al_2_Cl_7_] (1-butyl-3-methylimidazolium chloride), [C_4_mim][Ga_2_Cl_7_] and [C_4_mim][InCl_4_] as catalyst in toluene carbonylation [77].

## 3. Ga(III)-Based Ionic Liquids

Developing the topic of halometallate ILs, Yong et al. reported the application of [C_4_mim][GaCl_4_] as an active and efficient catalyst for the acetalization of aldehydes (Figure 1). In the case of conducting the reaction between methanol and benzaldehyde, it was possible to obtain the product with 81% yield in the presence of 5 mol% catalyst at room temperature. In comparison, the use of neat salt enables the yield of up to 55% after the same time and reaction conditions. Following the procedure, the authors also carried out the acetalization of acetaldehyde and propionaldehyde, coming up with corresponding acetals with 97% and 98% yields, respectively. In addition, the study proved that the catalyst could be used for at least five cycles without notable loss of efficiency [78].

Rangits et al. studied the performance of [C_4_mim][GaCl_4_] in hydroethoxycarbonylation of styrene (Figure 2). Tetrachlorogallate IL was used as a reaction medium and PdCl_2_(PPh_3_)_2_ as a catalyst. The ethoxycarbonylation of styrene led to two regioisomers, linear and branched esters. During the reaction in the presence of PdCl_2_(PPh_3_)_2_, the regioselectivity to branched ester achieved 77% and 67% yield, while the linear ester was obtained with 20% yield. After the addition of 1,1’-bis(diphenylphosphino)ferrocene (dppf) to a palladium catalyst, the regioselectivity of the process drastically changed. The branched ester was formed with 38% selectivity and 26% yield. On the other hand, in this catalytic system, the linear product yielded 43% [79].

Reports describing ILs containing a gallium atom and relatively large organic cations, such as 1-butyl-isoquinolinium gallium tetrachloride [(BIQL)GaCl_4_], are also known but without exemplary application [80].

Atkins et al. demonstrated the application of 1-ethyl-3-methylimidazolium heptachlorodigallate(III) [C_2_mim][Ga_2_Cl_7_] in the oligomerization process of linear olefins. Performed studies concerned 1-pentene transformation into a C20–C50 blend, which was used as a lubricant-based oil. Preliminary experiments revealed C20–C50 products with 6% conversion and 58% selectivity. Based on the initial data, the authors simulated the optimization of the conditions, according to which the selection of appropriate process parameters helped to achieve conversion of up to 61% and selectivity to the desired products at the level of 40% [81].

Alkylation is an important refinery process in the petroleum industry, which helps to produce large quantities of highly branched chain paraffin hydrocarbons. One of the most essential variants is the alkylation of light oil olefin (C3–C5) and isobutane, to obtain mainly isopentan and isooctan, which are mostly used as a blending stock for reformulated gasoline. Xing et al. applied a series of triethylammonium-based chlorogallate(III) ILs ([Et_3_NHCl]–GaCl_3_) with a molar fraction ratio of *χ*GaCl_3_ from 0.60 to 0.75. The system containing *χ*GaCl_3_ = 0.65 evaluated the highest product values among all investigated catalysts; therefore, it was next used with the addition of copper halide as a catalytic performance-improving additive [82,83]. As a result, it was possible to achieve better selectivity to C8 products up to 70.1 wt%, trimethylpentane up to 50.5 wt% and 91.3 wt% of total alkylate RON. Moreover, the research revealed the reusability of the catalytic system in at least nine cycles without any relevant loss in activity.

Among the possible applications, chlorogallate(III) Ils were also investigated as catalysts and media in alkylation of methyl linoleate with propene to prepare valuable branched high-molecular-weight products. Ga-based Ils were chosen due to their Lewis acid-catalytic ability and higher resistance to the presence of moisture than chloroaluminate Ils. The implementation of [C_4_mim][GaCl_4_/Ga_2_Cl_7_] and [BuIsoq][GaCl_4_/Ga_2_Cl_7_] shows different catalytic performances. In case of the use of [C_4_mim][GaCl_4_/Ga_2_Cl_7_], apart from the alkylation process, polypropene formation and oligomerization of methyl linoleate providing branched long-chain diesters and triesters was observed, whereas [BuIsoq][GaCl_4_/Ga_2_Cl_7_] was much more effective for propene oligomerization [84].

Oligomerization of 1-decene in the presence of a series of gallium(III) chloride and urea compositions was investigated to produce poly(*α*-olefins). Based on the data provided, an increase in the selectivity of the product was observed, due to the growing molar fraction of GaCl_3_ from 0.50 to 0.75 in the final composition of the catalyst [55].

Another application of gallium-containing Ils in the alkylation reaction can be found for [C_2_mim][Ga_2_Cl_7_]. In this Friedel–Crafts process, benzene and 1-decane were used as substrates for the model reaction. Among many catalytic options, the use of [C_2_mim][Ga_2_Cl_7_] helped to obtain phenyldecanes with a very high 91% yield and selectivity at room temperature with catalyst loading at 1 mol%. In traditional approaches, the described process requires higher temperatures, up to 70–150 °C [85].

In another approach, [C_2_mim]Cl-GaCl_3_ (*χ*GaCl_3_ = 0.50–0.75) was employed as a catalyst for Baeyer–Villiger oxidation. The use of hydrogen peroxide 30% aq. for 2-adamantanone oxidation resulted in a very fast transformation. Reported attempts reveal the influence of GaCl_3_ content on the catalyst activity. The use of [C_2_mim][Ga_3_Cl_10_] helped to reach 99% yield and [C_2_mim][Ga_2_Cl_7_] 93% yield after one minute of the reaction, while [C_2_mim][GaCl_4_] showed significantly less activity, and eventually, the yield reached 43% after 5 h [86].

An acid-catalysed Diels–Alder reaction (Figure 3) was performed in the presence of chlorometallate(III) ILs as catalysts. The presented research demonstrated the possibility of using ILs based on the 1-methyl-3-(triethoxysilylpropyl) imidazolium cation [tespmim]^+^ and chlorogalates anions [GaCl_4_]^−^, [Ga_2_Cl_7_]^−^, [Ga_3_Cl_10_]^−^. Furthermore, 5 mol% of catalyst [tespmim][Ga_2_Cl_7_] and [tespmim][Ga_3_Cl_10_] showed promising results, helping to achieved 73% and 98% conversions of methyl acrylate, respectively. The high yields of the product were accompanied by a very high stereoselectivity of the process (95:5 *endo*:*exo* ratio). The neutral [tespmim][GaCl_4_] did not show the catalytic properties (6% conversion, 80:20 *endo*:*exo* ratio) [87].

The studies were extended to explore ILs containing a borenium cation combined with 4-picoline (4pic), 1-methylimidazol (mim) or dimethylacetamide (dma) and [GaCl_4_]^−^ or [Ga_2_Cl_7_]^−^ as the counterion. For each of them, the maximum conversion of ethyl acrylate was achieved, despite that the less acidic ILs had to be used in larger amounts. However, full conversion was achieved in only 5 min and corresponded with an unchanged *endo*:*exo* (94:6) ratio by the diverse catalyst loading [67].

In 2019, Rogers et al. described a mixed-metal IL incorporating aluminium and gallium into the [Cl_2_GaOClAlClOGaCl_2_]^−^ anion. The mixed-metal species was created by combining triethylammonium chloride ([HN_222_]Cl), AlCl_3_, and GaCl_3_ in varied molar ratios ([HN_222_][*x*AlCl_3_ + (2 − *x*)GaCl_3_]Cl, where *x* = 2.0, 1.5, 1.33, 1.0, 0.67 or 0.5). Next, their activity was tested in Friedel–Crafts acylation and alkylation reactions (Figure 4). In both cases, the novel Al-Ga-based ILs enhanced the results and help to achieve a higher yield of products compared to the neat precursors and primary [HN_222_][Al_2_Cl_7_] or [HN_222_][Ga_2_Cl_7_]. Furthermore, 25.4% yield and up to 36.0% selectivity were achieved in acylation, whereas in the alkylation process, the obtained yields oscillated in 62–72% with the same level of selectivity [88].

Table 3 summarizes the applications of gallium(III)-based ILs as catalysts in organic synthesis.

## 4. In(III)-Based Ionic Liquids

The speciation of chloroindate(III) ionic liquids has been already described, including confirmation of the presence of [InCl_6_]^3−^, [InCl_5_]^2−^ and [InCl_4_]^−^ ions in compositions of ILs based on [C_4_mim]Cl and [C_8_mim]Cl in combination with InCl_3_ (0.25 ≤ *χ*InCl_3_ ≥ 0.75) [71].

One of the first works introducing indium-based ILs reveals their application as bifunctional catalysts performing as reaction media for Friedel–Crafts acylation reactions. The performance of the catalyst in benzoylation of anisole with benzoic anhydride was examined for [C_4_mim][InCl4] (χInCl_3_ = 0.67) and was compared with the use of InCl_3_ without a solvent, in 1,2-dichloroethane or with the ionic liquid 1-butyl-3-methylimidazolium bis(trifluoromethanesulfonyl)amide ([C_4_mim][NTf_2_]). The results turned out very valuable due to the very fast reaction progress for [C_4_mim][InCl_4_] used as a catalyst and reaction medium. Moreover, the benzoylation of benzene and it derivatives such as methoxybenzene, toluene, isobutylbenzene was performed with benzoic anhydride, benzoyl chloride or ethanoic anhydride (Figure 5) with 1.1–1.5 molar excess of the acylating agent. After 48 h of reaction, the yield for specific substrates oscillated around 81–97% with high 81–98% selectivity to the main products [90].

In another approach, indium(III)-containing ILs were studied in alcohol protection by transferring them to tetrahydropyranyl ethers. Herein, 1-alkyl-3-methylimidazolium tetrachloroindate(III) ILs were very efficient and recyclable catalysts. The tetrahydropyranylations reaction of various alcohols was performed using a microwave (MW) power unit for 5 min instead of conventional heating at 60 °C per 1 h. Carrying out the reaction in the presence of 25 mol% of [C_4_mim][InCl_4_] enabled improved output of the obtained ethers from phenol and benzyl alcohols from 20% and 39% yields to 88% and 85%, respectively. It was found that the impact of changing of the alkyl group attached to the imidazolium cation in IL was negligible in terms of catalytic activity in the model benzyl alcohol transition to ether. Additional studies proved not only high effectiveness of [C_4_mim][InCl_4_] in tetrahydropyranylation of cinnamyl alcohol, 1-phenylethanol, 2-phenylethanol, 3-methylpent-4-enol, but also impressive stability within five cycles without the loss of benzyl alcohol yield [91].

The R.S. Varma team further explored the perspective application of chloroindate(III) ILs and in the same year published the results of using [C_4_mim][InCl_4_] as a catalyst for the acetalization reaction of aromatic aldehydes with methanol. Replacing indium(III) chloride with 5 mol% of IL and carrying out the reaction at room temperature for 30 min helped to increase the yield of product from 40% to 70% [78].

The work by Varma et al. describes the series of haloindate(III)-based ionic liquids and their use as highly active catalysts for epoxides coupled with CO_2_ in order to produce cyclic carbonates (Figure 6). The catalytic efficiency was evaluated in propylene oxide coupling with CO_2_, conducting the process for 1 h in the presence of 5 mol% of the catalyst. The application of 1-butyl-3-methylimidazolium haloindate(III)-based ILs, such as [C_4_mim][InCl_4_], [C_4_mim][InCl_3_Br], [C_4_mim]InBr_3_Cl], [C_4_mim][InBr_4_] and [C_4_mim][InI_3_Cl] (all with *χ*InX_3_ = 0.5) helps to achieve up to 94% yield with full selectivity, whereas the usage of neat InCl_3_ did not initiate this reaction. Furthermore, changing the cation structure into 3-butyl-1,2-dimethylimidazolium (bdmim) resulted in increasing the yield to 95% and replacing it with the *n*-butylpyridinium (bPy) cation, enabling the highest yield of 97% [89].

Another report concerned the application of the same catalytic system established on [C_4_mim][InCl_4_] (*χ*InCl_3_ = 0.50) in the addition of allyltrimethylsilane, silyl enol ethers and ketene silyl acetals to cyclic *N*-acyliminium ions gain from carbamate *N*-Boc-2-methoxypyrrolidine (α-methoxycarbamate). Reactions carried out in chloroindium(III)-based IL helped to obtained product addition of allyltrimethylsilane and *N*-Boc-2-methoxypyrrolidinehigh with 80% yield, 68% yield for the addition of allyltrimethylsilane and piperidine derivative and a very high 89% yield for the allyltrimethylsilane and α-methoxytetrahydroisoquinoline reaction, whereas from silyl enol ethers and α-methoxycarbamate, it was possible to obtain products with 65–78% yields, and for ketene silyl acetal and various α-methoxycarbamate, the yields were in the range from 67% to 79% [92].

Gunaratne et al. proved that chloroindate(III) ILs are versatile and efficient catalysts for alkylation of phenol and their derivatives with alkenes. The studies employed various alkenes, such as propene, isobutene, diisobutene and 2-methylheptene. Interposed as a catalyst, In(III)-based IL was in the form of [C_4_mim][InCl_4_] (*χ*InCl_3_ = 0.67). The reaction of phenol with diisobutene and 2-methylheptene led to the main products, which were 4-substituted phenol derivatives with high 83% and 81% yields, respectively. However, with more reactive alkene, such as isobutene, the reaction was driven to the main product as disubstituted phenol (2,4-di-tert-butylphenol) with 78% yield, whereas the replacement of phenol with p-cresol in the reaction with isobutene led to as 78% yield to monosubstituted 2-tert-butyl-4-methylphenol. Furthermore, it turned out that all tested reactions with propene as the substrate were characterised with a non-selective distribution of products. The study was extended to explore the catechol alkylation with isobutene and diisobutene. The results showed equally high yields up to 85% and 88%, accordingly [93].

Another research focused on the synthesis of 3,4-dihydropyrimidinones in collaboration with [C_4_mim][InCl_4_] as an efficient catalyst. Conducted studies provided a series of 18 different Biginelli condensation products with very high yields (82–98%), proving the wide effectiveness of the IL [94].

The same catalytic system was examined for the addition of hydrazones to 1,3-diketones (Figure 7). [C_4_mim][InCl_4_] was applied in regioselective, one-pot condensation of aldehydes, arylhydrazines and acyclic or cyclic 1,3-diketones to synthesised pyrazoles. Performed attempts revealed an overwhelming advantage of In(III)-based IL, especially in the field of the regioselectivity of reaction between benzaldehyde, phenylhydrazine and 1-phenylbutane-1,3-dione, due to the formation of just one regioisomer. The highest achieved isolated yield was 86% with the presence of only one regio-product. Expanding the scope of the tested substrates to substituted arylhydrazine and various aldehydes, not only confirmed the effectiveness of the proposed catalytic system but also the universality of the method [95].

Several publications focused on biomass conversion into useful bio-based chemicals. Tiong et al. for the first time in 2017 presented the two-step method for conversions of oil palm empty fruit bunch (OPEFB) and mesocarp fibre (OPMF) biomass to levulinic acid and further esterification to ethyl levulinate [96]. The applied IL was described as eco-friendly, indium trichloride-1-methylimidazolium hydrogen sulphate, [Hmim][HSO_4_]-InCl_3_. In 2019, the authors again approached the topic and updated the study within the optimisation of operating parameters and critical comparison of the potential of the proposed methodology with different catalysts reported in the literature. During the experiments, parameters such as biomass depolymerization temperature (177 °C) and esterification temperature (105 °C) were selected; the first stage was carried out for 4.8 h, while the second stage took 12.2 h. The maximum yield of levulinic acid reached 17.7% (from OPEFB) and 18.4% (from OPMF), along with an efficiency of 74.1% and 86.4%, respectively. Additionally, in the esterification of levulinic acid, the ethyl ester was obtained with 18.7% yield (from OPEFB) and 20.1% (from OPMF), corresponding to the efficiency of 63.2% and 75.3% for OPEFB and OPMF. Apart from that, the catalytic system exhibited the capability to be recycled. Up to three following runs maintaining 75% efficiencies for both biomass transformations were performed [97]. The described applications of indium(III)-based ILs as catalysts are shown in Table 4.

## 5. Other Applications of Ga(III) and In(III)-Based Ionic Liquids

Chlorogallate(III) ILs were also frequently used for gallium electrodeposition [98,99]. Additionally, haloindate(III) ionic liquids have been reported as a precursor to indium(0) nanoparticle production through an electrochemical reduction [100].

The latest publication from 2021 studied ILs based on the tetrakis(pentafluoroethyl)gallate anion, [Ga(C_2_F_5_)_4_]^−^ [101]. These ILs with weakly coordinating anions were first presented for the first time in 2019, and the properties focusing on the potential electrochemical applications as conductive salts or additives to lithium-ion batteries were reported [102]. Gallate-based ILs contain various alkyl chain lengths at the imidazolium core, which is also combined with phenyl or phenyl derivatives and compared with [C_2_mim][Ga(C_2_F_5_)_4_] and [C_4_mim][Ga(C_2_F_5_)_4_]. New ILs featuring aryl-alkyl imidazolium cations combined with the tetrakis(pentafluoroethyl)gallate anion exhibited relatively low viscosities from 66 to 139 cP at 25 °C and exceptionally good electrochemical windows up to 5.15 V. On the other hand, the [C_2_mim]^+^ and [C_4_mim]^+^ based ILs showed very low viscosities of up to 29 cP at 25 °C but turned out to be electrochemically unstable [101].

A recently published paper provided information concerning gallium complexes created from methylidenemalonates and GaX_3_ with a strict 3/4 composition, which are postulated to be ILs. The object of the research, apart from the demonstration of interactions occurring in the complex, was the three-component addition to a triple bond that led to the generation of polyfunctional vinyl halides through Ga(III)-mediated halides [103].

This year, interesting research has appeared demonstrating the effective application of chlorogallate(III) IL in the process of desulfurization. Trihexyl(tetradecyl)phosphonium tetrachlorogallate, [P_66614_][GaCl_4_] was used as a catalyst for H_2_S conversion into sulphur. The reaction carried out by oxidative absorption of H_2_S in IL for 6 h at 100 °C enabled the removal of 92% of hydrogen sulphide. Other ILs tested by the authors which contained Fe(III) or Sn(II) showed even better performance, reaching conversions up to 97%. Nevertheless, the presented studies showed that ILs based on metal chlorides can also be efficient tools in gas purification processes [104].

The immobilization facilitates the efficient recycling of catalysts and enhances the stability of ILs deposited onto solid carriers [105]. In addition to the described above applications, ILs containing metals of group 13 (Al, Ga, In) are also successfully used as active phases immobilized on solid supports, such as amorphous silica, MCM-41, mesoporous organosilica SBA-15 or carbon nanotubes [51,61]. They form supported ionic-liquid phases (SILPs) and silica-based supported ionic liquid-like phase (SILLP)-type systems, which have recently become very popular as heterogeneous catalysts in various chemical processes, such as alkylated gasoline production [106], Diels–Alder cycloaddition [51,87], Friedel–Crafts reactions [107] and alkylation [108,109,110], isobutene oligomerization [111] or even the synthesis of *β*-keto enol ethers [112] and their use is the subject of numerous research and review publications [107,113,114].

## 6. Conclusions

Currently, it is crucial to focus on the replacement of conventional catalysts due to the development of environmentally and economically sustainable processes. Based on the literature results, in most cases, the unique properties of ILs helped not only to improve the performance of catalysts, yield and selectivity or to mitigate the reaction conditions, but also enabled easier and more economical separation of products from the reaction mixtures.

According to the reviewed gallium- and indium-based ILs, described in the literature systems include mainly halometallic anions incorporated into the IL structure. Considering the wide spectrum of possible applications of ILs, mostly in organic catalysis and the extraordinary and ambiguous properties of group 13 metals, the creation of new systems that could be built from other anionic units containing these metals appears to be extremely interesting and perspective.

## Figures and Tables

**Figure 1 molecules-28-01955-f001:**
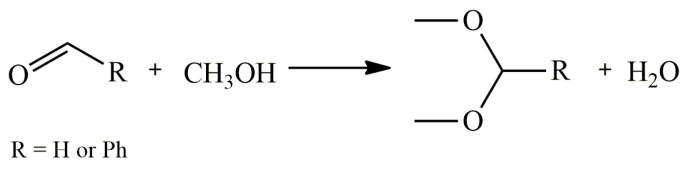
Acetalization of aldehydes with methanol.

**Figure 2 molecules-28-01955-f002:**
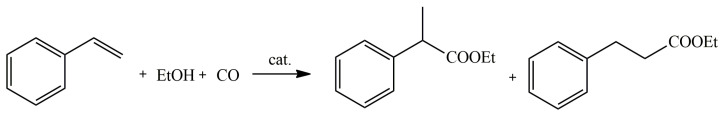
Hydroethoxycarbonylation of styrene.

**Figure 3 molecules-28-01955-f003:**
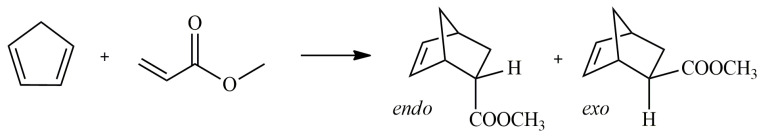
Diels–Alder reaction of cyclopentadiene and methyl acrylate.

**Figure 4 molecules-28-01955-f004:**
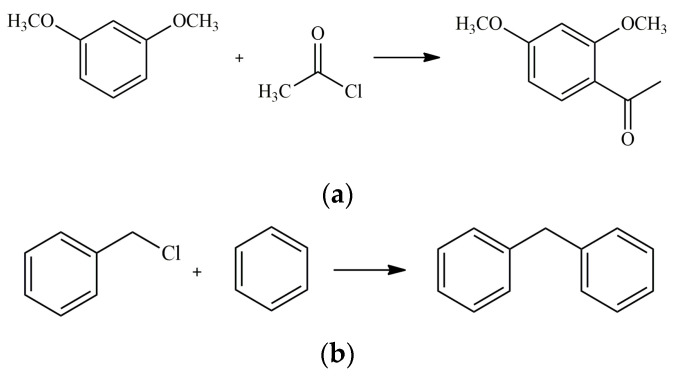
Friedel–Crafts acylation and alkylation reactions as: (**a**) acylation of 1,3-dimethoxybenzene and acetyl chloride; (**b**) alkylation of benzene with benzyl chloride.

**Figure 5 molecules-28-01955-f005:**
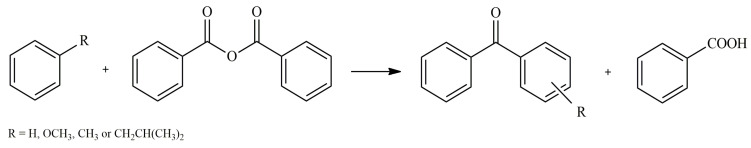
Benzoylation of methoxybenzene by benzoic anhydride.

**Figure 6 molecules-28-01955-f006:**
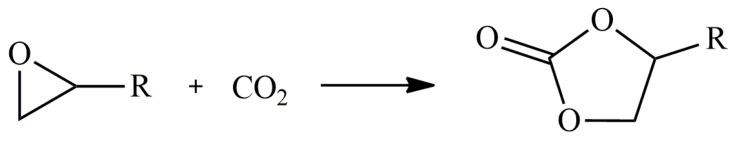
Epoxides coupling with CO2 to synthetized cyclic carbonates.

**Figure 7 molecules-28-01955-f007:**
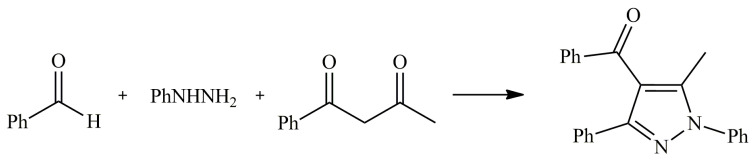
Synthesis of pyrazole by aldehyde, hydrazone and 1,3-diketone condensation.

**Table 1 molecules-28-01955-t001:** Acceptor numbers of common substances [59].

Substance	AN
hexane	0
pyridine	14.2
methanol	41.3
ethanoic acid	52.9
water	54.8
trifluoroethanoic acid	105.5
methanesulphonic acid	126.1
trifluoromethanesulphonic acid	129.1

**Table 2 molecules-28-01955-t002:** Series of anionic spicies and acceptor numbers for selected chlorometallate(III) ionic liquids based on [C_8_mim]^+^ cation [38,39,41].

IL	Anionic Speciation	AN
*χ*AlCl_3_ = 0.33	Cl^−^, [AlCl_4_]^−^	93.2
*χ*AlCl_3_ = 0.50	[AlCl_4_]^−^	91.8
*χ*AlCl_3_ = 0.67	[Al_2_Cl_7_]^−^	96.0
*χ*GaCl_3_ = 0.33	[GaCl_4_]^−^	21.7
*χ*GaCl_3_ = 0.50	[GaCl_4_]^−^	45.9
*χ*GaCl_3_ = 0.67	[Ga_2_Cl_7_]^−^	99.5
*χ*GaCl_3_ = 0.75	[Ga_3_Cl_10_]^−^	107.5
*χ*InCl_3_ = 0.25	[InCl_6_]^3−^	32.5
*χ*InCl_3_ = 0.50	[InCl_4_]^−^	57.1
*χ*InCl_3_ = 0.67	[InCl_4_]^−^	58.4

**Table 3 molecules-28-01955-t003:** Summary of reactions catalysed by gallium(III)-based ILs.

Process	Substrates	Ionic Liquid	Reaction Conditions	Performance	Ref.
Acetalization	methanol, aldehydes	[C_4_mim][GaCl_4_] (*χ*GaCl_3_ = 0.50)	aldehyde 5.66 mmol; catalyst 5 mol%; 30 min; rt	81% yield from benzaldehyde	[78]
				97% yield from acetaldehyde	
				98% yield from propionaldehyde	
Ethoxycarbonylation	styrene, ethanol, CO	[C_4_mim][GaCl_4_] (*χ*GaCl_3_ = 0.50)	catalyst PdCl_2_(PPh_3_)_2_	67% yield; 77% selectivity	[79]
Oligomerization	1-pentene	[C_2_mim][Ga_2_Cl_7_] (*χ*GaCl_3_ = 0.67)	catalyst 0.10–0.45 mol%; 1 h; 0–20 °C	6% conversion; 58% selectivity to C20–C50 blend	[81]
	1-decene	Urea-GaCl_3_ (*χ*GaCl_3_ = 0.50–0.75)	catalyst 1 wt%; 1 h; 120 °C	71.5–78.5% conversion; 34.1–38.7% selectivity to C20	[55]
Alkylation	isobutane, olefin (C3–C5)	[Et_3_NHCl]-GaCl_3_ (*χ*GaCl_3_ = 0.65)	40 mL hydrocarbon feed; molar ratio of isobutane/butene 10:1; catalyst/hydrocarbon feed ratio 0.4; CuCl = 5 mol%; 5 bar; 15 min; 15 °C.	selectivity of C8 products up to 70.1 wt%, trimethylpentane up to 50.5 wt% and total alkylate RON were 91.3 wt%	[82]
		[Et_3_NHCl]-GaCl_3_/CuCl (*χ*GaCl_3_ = 0.65)			
	methyl linoleate, propene	[C_4_mim][GaCl_4_/Ga_2_Cl_7_] (*χ*GaCl_3_ = 0.60);	methyl linoleate 11 mmol; propane 7 bar; 8 h; 100 °C	no data	[84]
		[BuIsoq][GaCl_4_/Ga_2_Cl_7_] (*χ*GaCl_3_ = 0.60)			
Baeyer–Villiger oxidation	2-adamantanone, H_2_O_2_	[C_2_mim]Cl-GaCl_3_ (*χ*GaCl_3_ = 0.67–0.75)	2-adamantanone 0.67 mmol; H_2_O_2_ 30% aq. 1.34 mmol; catalyst 100 mol%; 1 min; rt	93–99% yield	[86]
Diels–Alder	cyclopentadiene, methyl acrylate	[tespmim][GaCl_4_] (*χ*GaCl_3_ = 0.50)	cyclopentadiene 4 mmol; methyl acrylate 6 mmol; catalyst 5 mol%; 5 min; 25 °C	6% conversion; 80:20 *endo*:*exo* products ratio	[87]
		[tespmim][Ga_2_Cl_7_] (*χ*GaCl_3_ = 0.67)		73% conversion; 95:5 *endo*:*exo* products ratio	
		[tespmim][Ga_3_Cl_10_] (*χ*GaCl_3_ = 0.75)		98% conversion; 95:5 *endo*:*exo* products ratio	
	cyclopentadiene, ethyl acrylate	[BCl_2_(4pic)][GaCl_4_] (*χ*GaCl_3_ = 0.50) [BCl_2_(4pic)][Ga_2_Cl_7_] (*χ*GaCl_3_ = 0.67) [BCl_2_(dma)][Ga_2_Cl_7_] (*χ*GaCl_3_ = 0.67) [BCl_2_(mim)][Ga_2_Cl_7_] (*χ*GaCl_3_ = 0.67)	cyclopentadiene 24 mmol; ethyl acrylate 16 mmol; catalyst 0.1–0.5 mol%; 5 min; 0 °C	100% conversion; 94:6 *endo*:*exo* products ratio	[67]
Friedel–Crafts alkylation	1-decene, benzene	[emim][Ga_2_Cl_7_] (*χ*GaCl_3_ = 0.67)	1-decene 128.34 mmol; benzene 898.27 mmol; catalyst 1 mol%; 1,5 h; 20 °C	91% yield and 91% selectivity to all of 2,3,4,5-phenyldecanes	[85]
				40% yield and 36% selectivity to 2-phenyldecane	
	benzyl chloride, benzene	[HN_222_][Ga_2_Cl_7_] (*χ*GaCl_3_ = 0.67)	benzyl chloride 1 mmol; benzene 5 mL; catalyst 10 mol%; 15 min; 30 °C	57.8% yield; 57.8% selectivity	[88]
		[HN_222_][*x*AlCl_3_ + (2 − *x*)GaCl_3_]Cl (*x* = 0.5–1.5)		62.5–72.0% yield; 62.5–72.0% selectivity	
Friedel–Crafts acylation	1,3-dimethoxybenene, acetyl chloride	[HN_222_][Ga_2_Cl_7_] (*χ*GaCl_3_ = 0.67)	1,3-dimethoxybenene 1 mmol; acetyl chloride 1 mmol; catalyst 10 mol%; 3 h; 30 °C	16.6% yield; 33.7% selectivity	
		[HN_222_][*x*AlCl_3_ + (2 − *x*)GaCl_3_]Cl (*x* = 0.5–1.5)		13.7–25.4% yield; 20.9–36.0% selectivity	
Epoxides coupling	propylene oxide, CO_2_	[C_4_mim][GaCl_4_] (*χ*GaCl_3_ = 0.50)	propylene oxide 51.6 mmol; catalyst 0.5 mol%; CO_2_ 100 psi; 1 h; 120 °C	17% yield; 100% selectivity	[89]

**Table 4 molecules-28-01955-t004:** Summary of reactions catalysed by indium(III)-based ILs.

Process	Substrates	Ionic Liquid	Reaction Conditions	Performance	Refs.
Fiedel–Crafts acylation	benzene derivatives with benzoic anhydride (BA), benzoyl chloride (BC) or ethanoic anhydride (EA)	[C_4_mim][InCl_4_] (*χ*InCl_3_ = 0.67)	anisole 113 mmol; anhydride 124 mmol; catalyst 10 mol%; 48 h; 80–120 °C	97% yield and 98% selectivity for anisole and BA	[90]
				89% yield and 98% selectivity for anisole and EA	
				81% yield for benzene and BC	
				86% yield and 81% selectivity for toluene and BA	
				87% yield and 86% selectivity for isobutylbenzene and BA	
				96% yield and 83% selectivity for isobutylbenzene and BC	
Tetrahydropyranylation of alcohols	3,4-dihydropyran, alcohol	[C_4_mim][InCl_4_] (*χ*InCl_3_ = 0.50)	3,4-dihydropyran 11 mmol; alcohol 10 mmol; catalyst 25 mol%; 5 min; 100 W of MW irradiation	88% yield from phenol	[91]
				85% yield from benzyl alcohol	
				84% yield from cinnamyl alcohol	
				86% yield from 1-phenylethanol	
Acetalization	benzaldehyde, methanol	[C_4_mim][InCl_4_] (*χ*InCl_3_ = 0.50)	benzaldehyde 5.66 mmol; catalyst 5 mol%; rt for 30 min	70% yield	[78]
Epoxides coupling	propylene oxide, CO_2_	[C_4_mim][InCl_4_], [C_4_mim][InCl_3_Br], [C_4_mim]InBr_3_Cl], [C_4_mim][InBr_4_], [bPy][InCl_4_], (*χ*InCl_3_ = 0.50)	propylene oxide 51.6 mmol; catalyst 0.5 mol%; CO_2_ pressure 100 psi; 1 h; 120 °C	92–97% yields; 100% selectivity	[89]
Nucleophile additions to cyclic *N*-acyliminium ions	α-methoxycarbamate and nucleophile	[C_4_mim][InCl_4_] (*χ*InCl_3_ = 0.50)	α-methoxycarbamate 0.25 mmol; nucleophile 0.38–0.50 mmol; catalyst 0.1 mL; rt; 24 h	80–89% isolated yield from allyltrimethylsilane	[92]
				76–78% isolated yield from silyl enol ethers	
				77–79% isolated yield from ketene silyl acetal	
Alkylation	phenol, p-cresol or catechol with isobutene, diisobutene or 2-methylheptene	[C_4_mim][InCl_4_] (*χ*InCl_3_ = 0.67)	phenol to indium mole ratio = 50:1; 100–110 °C	78–88% yields	[93]
Biginelli condensation	benzaldehyde, ethylacetoacetate and urea	[C_4_mim][InCl_4_] (*χ*InCl_3_ = 0.50)	aldehyde 5 mmol; *β*-dicarbonyl compound 5 mmol; urea 5 mmol; catalyst 0.5 mL; 25-55 min; 50 °C	98% yield	[94]
	aldehyde, *β*-dicarbonyl compound and urea			82–97% yield	
Condensation of aldehydes, hydrazones and 1,3-diketones	aldehyde, arylhydrazine and 1,3-diketones	[C_4_mim][InCl_4_]	aldehyde 1 mmol; arylhydrazine 1 mmol; 1,3-diketone 1.2 mmol; catalyst 1.2 mmol; 1–2 h; 140 °C	61–90% yields; 100% regioselectivity	[95]
Biomass depolymerisation to levulinic acid	oil palm empty fruit bunch (OPEFB) and mesocarp fiber (OPMF) biomass, ethanol	[Hmim][HSO_4_]-InCl_3_	0.15 mmol of InCl_3_ in IL; 6.6:1 (*w*/*w*) of ILs to biomass; 22.7% (*w*/*w*) of H_2_O; 177 °C; 4.8 h	17.7% (from OPEFB) and 18.4% (from OPMF) yields to LA; 74.1% (from OPEFB) and 86.4% (from OPMF) efficiencies	[96,97]
Levulinic acid esterification to ethyl levulinate			7.2:1 (*v*/*v*) of ethanol to LA ratio; 105 °C; 12.2 h	18.7% (from OPEFB) and 20.1% (from OPMF) yields to EL; 63.2% (from OPEFB) and 75.3% (from OPMF) efficiencies	

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
