# Peer review of "Gallium(III)- and Indium(III)-Containing Ionic Liquids as Highly Active Catalysts in Organic Synthesis"

_molecules, 2023, doi:10.3390/molecules28041955_

Round 1

Reviewer 1 Report

Ionic liquids are widely used in chemistry and chemical industry because of their unique properties.In the background section, the author emphasized its important role in the field of catalysis. This is necessary and meaningful. However, in addition to Ga metal you mentioned, more noble metals such as Cu should also be mentioned to meet the requirements of readers in this field (Green Energy & Environment, 2022. Doi:10.1016/j.gee.2022.01.005).The logic of the full text is very clear, but some necessary details can be improved to improve the quality and influence of the manuscript.It is suggested that the evolution of physicochemical properties due to different metal concentrations in ionic liquids can be plotted as a phase diagram?In addition, there is less discussion about the mechanism in this paper, such as the potential influence of H atom at C2 position?In addition, the dissolution mechanism of substrate in ionic liquids is suggested to be discussed more deeply.Is there a lack of application of ionic liquids on carbon based carriers?

Author Response

Dear Reviewer,

Reviewer 2 Report

In the proposed reviews, the authors presents the sued of Ga and In containing IL to catalyze common organic reactions with better yield, selectivity. The manuscript is very well organized and written. 

However, the synthesis and characterization of these families of ionic liquids are not mentionned and thus, a brief description on different approaches for synthesis and characterization is highly recommended (Raman, etc.)

Author Response

Dear Reviewer,

Thank you very much for your comments. We have restructured the manuscript to address all your remarks. We are resubmitting the corrected version. Please find the replies to your questions listed below in a point-by-point fashion.

With best regards,

Anna Chrobok

Q1: However, the synthesis and characterization of these families of ionic liquids are not mentionned and thus, a brief description on different approaches for synthesis and characterization is highly recommended (Raman, etc.)

A1: Thank you for the comment. The synthesis and characterization of the ionic liquids was discussed as follows: ” The ionic liquids described in the review are generally synthesized via mixing the organic halide salt with the appropriate metal halide in a certain molar ratio [38]. Afterwards, the speciation involves physicochemical characterization by a wide range of technics [38,41,72–74] such as multinuclear NMR spectroscopy, infrared spectroscopy or Raman spectroscopy, mass spectrometry, sometimes X-ray spectroscopy (XPS or XAS) and UV-VIS spectroscopy. To determine the thermal properties of ionic liquids differential scanning calorimetry or thermogravimetric analysis have to be performed. Subsequently crucial for this group of ILs is their acidity determined based on acceptor properties, described in the next section [39,45]. The individual properties of metal-based ILs depend not only on the metal character but also on the form of cation and anion formed and hence the interionic interactions [75,76]. The molar ratio in which substrates were used also plays a significant role, especially taking into consideration the acidity of the system. The halometallate ILs comprising group 13 metals, mostly in the form of chloride were summarized to exhibit catalytic ability resulting from the anion speciation. The catalytic properties outcome directly from the anionic polynuclear halometallate species formation ([MxCly]z ). In chloroaluminate(III) ILs anions like Cl-, [AlCl4]-, [Al2Cl7]-, [Al3Cl10]- and [Al4Cl13]- exist. In the chlorogallate(III) systems analogue anions were found such as Cl-, [GaCl4]-, [Ga2Cl7]- and [Ga3Cl10]-. While in chloroindate(III) ionic liquids only [InCl6]3-, [InCl5]2- and [InCl4]- anions were reported. The occurrence of various anionic species is dependent on metal chloride molar fraction (χMCl3) in IL composition, as shown in Table 2 [41,71]. Regarding the physical state of halometallate ionic liquids, it can also be divided based on different metal concentrations in ILs mixtures. Chloroaluminates(III) create homogenous ionic liquids in compositions where χAlCl3 is less than or equal to 0.67. However, for higher values of χAlCl3, aluminum(III) chloride was precipitated. In contrast, chlorogallate(III) ILs form homogenous liquids across the entire range studied χGaCl3 from 0.25 to 0.75. When it comes to chloroindate(III) ILs, clear ionic liquids were only observed when χInCl3 was less than or equal to 0.50, while beyond this composition the mixtures were biphasic with solid particles suspended within the ionic liquid [45,72,73].”

Reviewer 3 Report

The article is discussing the catalytic activity of ionic liquids composed of gallium and indium chlorometalates. While the thalium representatives of group III are scarce, little is said about borium and Al containing ionic liquids. Moreover, little is said about the uniqueness of group III halometallates compared to other groups. The way of classification of reaction mechanisms is well done, and the citations are proper except the fact that more ionic liquids devoted books could be cited in the introductory part. Reference 29 is lacking.

Author Response

Dear Reviewer,

Thank you very much for your comments. We have restructured the manuscript to address all your remarks. We are resubmitting the corrected version. Please find the replies to your questions listed below in a point-by-point fashion.

With best regards,

Anna Chrobok

Q1: While the thalium representatives of group III are scarce, little is said about borium and Al containing ionic liquids.

A1: In the introduction section, the following sentences were added to highlight the boron-containing ionic liquids and their use applications: “In group 13 of the periodic table, there is also boron representing semimetals. Boron-based ionic liquids are a subgroup of ILs, which were investigated for a range of applications, including catalysis, energy storage and conversion, and separation processes [65–69]. Despite the potential benefits, there are also some challenges associated with the use of boron-based ionic liquids like their high viscosity and others nevertheless, ongoing research efforts aim to overcome these limitations and further expand the use of boron-based ionic liquids in various fields [70].” Additionally, the influence of metal halide concentration in ionic liquids on physicochemical properties of ILs (including Al-based ILs) was discussed together with the description of possible anionic species present in metal-based ionic liquids. 

Q2: Moreover, little is said about the uniqueness of group III halometallates compared to other groups.

A2: The relevant changes have been made, highlighting the group III halometallates.

Q3: The way of classification of reaction mechanisms is well done, and the citations are proper except the fact that more ionic liquids devoted books could be cited in the introductory part.

A3: The relevant books on ionic liquids have been added (ref. 4, 5, 7, 8 and 106).

Q4: Reference 29 is lacking.

A4: It was corrected, now as a Ref. 38.

Round 2

Reviewer 1 Report

Accepted.